# Effect of Fabric Areal Weight on the Mechanical Properties of Composite Laminates in Carbon-Fiber-Reinforced Polymers

Marina Andreozzi, Iacopo Bianchi , Serena Gentili, Tommaso Mancia and Michela Simoncini *

Department of Industrial Engineering and Mathematical Sciences, Università Politecnica delle Marche,
Via Brecce Bianche 12, 60131 Ancona, Italy; m.andreozzi@pm.univpm.it (M.A.); i.bianchi@pm.univpm.it (I.B.);
s.gentili@pm.univpm.it (S.G.); t.mancia@staff.univpm.it (T.M.)
* Correspondence: m.simoncini@staff.univpm.it; Tel.: +39-071-2204443

**Abstract:** The present work aims at studying the effect of the reinforcing fabric areal weight on the mechanical properties of composite laminates in carbon-fiber-reinforced polymers. Three different pre-impregnated $2 \times 2$ twill weaves, characterized by the different areal weight values of 380, 630, and 800 g/m$^2$ were used to produce laminates. These areal weights were selected to represent typical values used in structural application. A hand lay-up technique followed by an autoclave cycle curing was employed to produce the laminates. The desired final thickness of the laminates was obtained by laying-up a different ply number, as a function of the areal weight and thickness of each fabric. Uniaxial tensile and in-plane shear response tests were performed on samples obtained from laminates after curing. Furthermore, the presence of voids in composite materials were detected by performing resin digestion tests. Finally, light optical microscopy and stereomicroscopy analyses allowed observing the different arrangement of the plies in the cross-sections of laminates after curing and evaluating the degree of compaction as a function of the reinforcing fabric used. It was demonstrated that the fabric areal weight significantly affects the mechanical performances of the composite laminates; specifically, the decrease in the areal weight of the twill weave leads to an increase in tensile strength, elastic modulus, and in-plane shear stress, i.e., of about 56.9%, 26.6%, and 55.4%, respectively, if 380 g/m$^2$ and 800 g/m$^2$ fabrics are compared. These results are crucial for an optimal material selection during the design process for industrial applications and help to better understand composite material behavior.

**Keywords:** CFRP; fabric areal weight; mechanical performances; tensile test; in-plane shear tests

## 1. Introduction

Composite materials are well known for their extraordinary characteristics in terms of specific mechanical performances, especially when continuous carbon fibers are used as reinforcement. For such reasons, Carbon-Fiber-Reinforced Polymers (CFRPs) exhibit constant growth in applications where lightness and strength are the main targets to be fulfilled [1–7]. In order to create strong composite structures, multilayer laminates in CFRP can be used, obtained by laying up different pre-impregnated layers until the desired thickness is reached. The layers deposition must be accurately performed in order to reach the desired performances. The lay-up process can be performed manually [8,9] or by means of automated systems [10]. Depending on the layer arrangement, different types of laminates can be obtained. The maximum mechanical properties are obtained by manufacturing unidirectional composites, in which the continuous carbon fibers are parallelly oriented and in the same direction [11]. Unfortunately, this configuration can be used in very few applications because the properties are optimized only in the longitudinal direction of the fibers. Therefore, it is not an optimal configuration for applications in which combined loadings are applied along different directions. To overcome such drawbacks, cross-ply laminates, in which fibers are oriented at 0° and 90°, can be stacked since they can provide

excellent mechanical performances in more than one direction. Furthermore, a stacking sequence of 0°, 45°, 90°, −45°, 0° can be used to make the material more orthotropic, even though the high mechanical properties typical of the unidirectional laminates cannot be achieved [11]. In addition, fibers can also be oriented along two perpendicular directions obtaining bidirectional fabrics. They are characterized by a warp and weft, in which the weave type determines how often the transverse (0/90°) strands of fibers in fabrics are mutually interleaved. Composite woven fabrics offer many advantages in terms of deformation capability, dimensional stability, good conformability, damage tolerance, easy handling, and low fabrication costs as compared to unwoven-fabric composites [12]. Different architectures can be manufactured, such as plain, satin, and twill weaves. The different fiber architectures can significantly affect the mechanical properties of the fiber-reinforced composites. For this reason, the type of fabric reinforcement must be carefully chosen, considering the application and the mechanical stresses that the part will have to withstand [13]. Unfortunately, woven fabric composites are characterized by drawbacks leading to a reduction in the mechanical properties as compared to unidirectional laminates. Specifically, the fibers undulation, or crimping, causes a misalignment of the fibers outside the fabric plane, and a consequent decrease in both stiffness and strength of the composite materials [14–18]. As a matter of fact, failure of fabric composites generally begins at lower strain values than those at which unidirectional composites fail. Earlier crack initiation in textile composites does not immediately result in catastrophic failure, but it can affect the component performance. Many researchers have dealt with these issues to evaluate the different damage methods as a function of the process and material parameters [19–21]. Specifically, it was noted that the areal weight (AW) of the fabric, that is the amount (mass) of fibers within an area unit of the fabric, is a fundamental parameter that affects the fabric failure mechanisms, since laminates with a low AW are characterized by a failure involving fibers, while a laminate with a high AW value undergoes delamination. Mateusz Koziol [13] demonstrated that the increase in AW leads to an increase both in tensile strength and elasticity modulus, and a decrease in the deformability of laminates reinforced with different fiber typologies. However, only 200 g/m$^2$ carbon fabric laminates were analyzed. Di Bella et al. [22] studied the effect of the AW on the mechanical properties of bidirectional-flax-fabrics-reinforced composites; they demonstrated that the flexural properties decrease when the AW increases because of the delamination effect. Recently thin-ply composites have received an increasing attention due to their high mechanical performances [23,24]. Unfortunately, few studies concerning the effect of fabric AW on the mechanical properties of composite laminates in carbon-fiber-reinforced polymers are available in scientific literature.

In this framework, the present investigation aims at studying the influence of fabric AW on the performances of laminates in carbon-fiber-reinforced polymers. To this purpose, three different pre-impregnated 2 × 2 twill weaves in CFRP, characterized by different AW values of 380 g/m$^2$, 630 g/m$^2$, and 800 g/m$^2$ and supplied by company operating in automotive field, were used. These fabrics are the most employed in the automotive sector, specifically in manufacturing of structural components, because they are characterized by high drapability and limited crimping. In order to obtain the desired final thickness of laminates, a different number of plies was laid-up, i.e., three layers of fabric with the highest AW investigated, four layers with the 630 g/m$^2$ weave, and six plies of fabric with the lowest AW. Samples were cut from the composite laminates after curing. The performances of the composites were evaluated by means of void quantities analysis, uniaxial tensile, and in-plane shear tests to identify the configuration that led to optimal mechanical performance. The effect of the fabric AW on the response of the CFRP laminates was investigated.

The paper is organized as follows: after the introduction, Section 2 presents the description of the materials investigated and the methods applied. In Section 3, the results are shown and discussed. Finally, the main conclusions are summarized.

## 2. Materials and Methods

### 2.1. Materials

Three different pre-impregnated fabrics in carbon-fiber-reinforced composite, characterized by a 2 × 2 twill weave, were used in the present manuscript. These materials were supplied by an Italian company leader in automotive and motorsport fields and were selected since largely employed in several structural applications. The carbon fibers in twill weave architecture were impregnated using the E3 150N thermosetting epoxy resin, suitable for autoclave curing. The carbon fiber weaves were characterized by different AWs (AWs), equal to 380, 630, and 800 g/m$^2$. The composite fabrics characterized by the values of 380 and 630 g/m$^2$ were obtained using a fiber size of 12K whilst the fabric with the AW equal to 800 g/m$^2$ was realized using a fiber size of 24K.

### 2.2. Fabrication of Composite Laminates

The plies were cut from the carbon fabric rolls before the deposition process. After a treatment of the mold surface by using a release antiadhesive agent to avoid the polymer sticking to the surface, the lamination of fabric plies on the surface of the mold was performed by means of the hand lay-up technique. A fabric layer was first placed by hand onto the mold surface and subsequent layers were placed on the first one until the required thickness was reached. A partial removal of the voids was performed by vacuum between successive deposition operations of CFRP twill fabrics in order to obtain effective compaction and to eliminate the air between the layers of material and between the material and the mold. The presence of air inside the laminate reduces its mechanical performance [6]. Once the plies were stacked, a vacuum bag was applied, and an autoclave cycle curing was performed. The curing cycle required the support of temperature and vacuum to achieve both the correct viscosity level of the resin for its uniform distribution through the fibers and the elimination of trapped air. Table 1 summarizes the process parameters used for the polymerization cycle in autoclave.

**Table 1.** Autoclave cure process parameters.

| | |
|---|---|
| Vacuum bag pressure | 0.9 bar |
| Autoclave pressure | 4.0–7.0 bar |
| Ramp rate | 1–5 °C/min |
| Cure cycle | 90 min at 130 °C (+5/−0 °C) |
| Cool rate | 2–3 °C/min until 60 °C |

A final laminate thickness of 2.5 mm was obtained using the different fabrics. Since the thickness of CFRPs plies is related to the fabric AW, in order to reach the desired laminate thickness, different layups were employed according to the AW of each ply. Specifically, six plies of the fabric characterized by the lowest AW of 380 g/m$^2$, four plies of the fabric with AW of 630 g/m$^2$, and three plies of the 800 g/m$^2$ AW fabric were used.

A typical panel in CFRP composite, obtained by means of hand lay-up process, is shown in Figure 1.

### 2.3. Chemical and Mechanical Tests

The performances of the composites were evaluated by means of void quantities analysis, uniaxial tensile, and in-plane shear response tests. The effect of the AW of twill weaves on the mechanical response of the CFRP laminates was investigated.

#### 2.3.1. Resin Digestion Tests and Void Quantity Analysis

The test to detect the presence of voids in composite materials was performed following the ASTM D3171 standard according to procedure B: "matrix digestion using sulfuric acid/hydrogen peroxide". The resin digestion test allows to determine the reinforcement and matrix content (by weight or volume) as well as void volume. Figure 2 shows the dry fibers after the resin digestion.

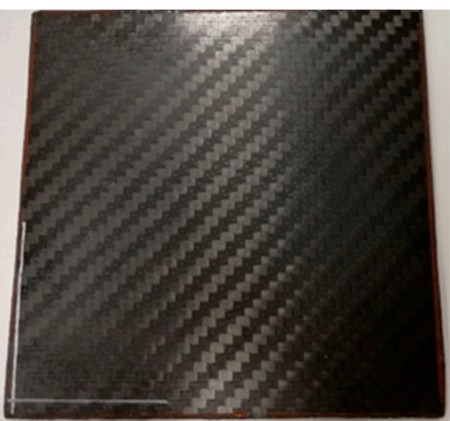

**Figure 1.** A flat laminate in CFRP produced by hand lay-up process, after curing.

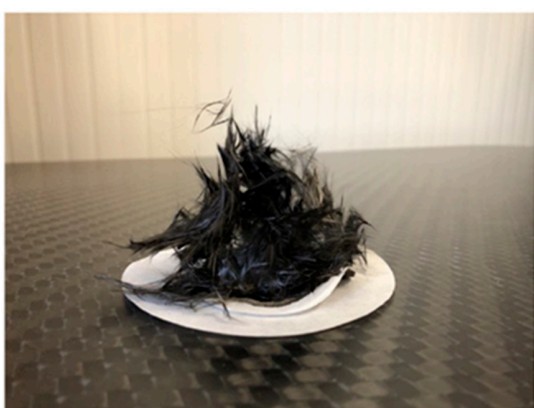

**Figure 2.** Dry fibers after resin digestion the void quantity analysis.

### 2.3.2. Mechanical Tests

Uniaxial tensile and in-plane shear response tests were carried out at room temperature, according to the ASTM D3039 and ASTM D3518 standards, respectively. These standard tests were chosen as they allow to obtain the main mechanical properties of CFRP composites, such as ultimate tensile strength, elastic modulus, and maximum in-plane shear stress, often used as a performance index during the design phase of structural components. Samples were obtained from the composite panels, using the servo-hydraulic testing machine MTS 810® (MTS Systems Corporation, Eden Prairie, MN, USA), equipped with a load cell of 250 kN, at a constant crosshead speed of 2 mm/min. The samples, with length, width, and thickness equal to 250 mm, 25 mm, and 2.5 mm, respectively, were water-jet cut from the cured laminates in order to obtain the carbon fiber directions in the weaves at $0°/90°$ (for the tensile tests) and $\pm45°$ (for the in-plane shear response tests) with respect to the loading direction. In order to reduce stress concentrations that can be caused by the clamping system of the testing machine and to avoid premature failure, tabs in glass fiber composite material were bonded to each end of the tensile specimens using two components of epoxy adhesive. Each sample was instrumented by applying strain gages in order to acquire the instantaneous strain along the loading direction during test. The results obtained by tensile and in-plane shear tests, in terms of the load and strain, allowed to plot tensile stress vs. tensile strain curves and shear stress vs. shear strain curves, by which the maximum values of strength were obtained.

In order to compare the mechanical properties of composite laminates obtained using fabrics with different AWs, a normalization process with respect to the fiber volume fraction was performed. As a matter of fact, the variation of fiber content in the composite laminate leads to significant changes in the mechanical properties of the laminates. Therefore, the results obtained from the mechanical tests were normalized to a quantity of fiber volume

fraction equal to 60% by taking into account the results of the resin digestion according to the guidelines for the characterization of structural composite materials [25]. At least five tensile and in-plane shear tests were performed for each experimental condition investigated in order to guarantee the repeatability of the results.

### 2.3.3. Light Optical Microscopy and Stereomicroscopy

The cross-sections of laminates after curing were observed using a light optical microscope Leica DMi8 (Leica Microsystems GmbH, Wetzlar, Germany) in order to observe the different arrangement of the plies in section and evaluate the degree of compaction as a function of the fabric weave thickness.

Furthermore, in order to investigate the failure mode of specimens subjected to tensile and in-plane shear tests, a Leica EZ 4D stereomicroscope was used at different magnification levels to observe the fractured samples along the loading direction.

## 3. Results and Discussion

Figure 3 shows the light optical microscopies of the longitudinal sections and external flat surfaces of composite laminates after curing obtained by laying-up 2 × 2 twill weaves with different AW values. It can be observed that weaving necessarily introduces a fiber misalignment (crimp) inside the composite that can affect the mechanical performances of laminates and the failure mechanisms. Irrespective of the AW taken into account, the twill architectures are characterized by a quasi-elliptical shape. Furthermore, it can be seen that the composite laminates are manufactured using a lower number of fabrics layers with increasing the AW factor to obtain the desired final thickness of composite laminate equal to 2.5 mm. A more lenticular fiber shape appears as the weave with the lowest AW value is considered (Figure 3a) as compared to the other two fabrics with an AW of 630 g/m$^2$ and 800 g/m$^2$ (Figure 3b,c), which show different stacking patterns.

The presence of voids into laminates strongly reduces the material strength. Moreover, voids affect the humidity absorption and other properties such as chemical and physical degradation of fibers. As far as the evaluation of the reinforcement, resin, and void contents is concerned, results of the resin digestion test are shown in Table 2. Values are reported as a percentage by weight (%$w/w$) or by volume (%$v/v$).

**Table 2.** Void content in the composite laminates obtained using twill fabric.

| AW [g/m$^2$] | Resin Content [%$w/w$] | Fiber Content [%$v/v$] | Void Volume Percentage [%$v/v$] |
|---|---|---|---|
| 380 | 43.9 | 46.0 | 0.10 |
| 630 | 32.1 | 58.3 | 0.35 |
| 800 | 29.7 | 61.1 | 0.15 |

It can be seen that the fabric characterized by the lowest AW value shows the lowest void content in percentage, demonstrating that the lightest weave can be subjected to a more efficient compaction operation, while the fabric with an AW of 630 g/m$^2$ presents the higher percentage of voids.

In order to evaluate the effect of the fabric AW on the mechanical performances of CFRP composite laminates, tensile and in-plane shear response tests were carried out at room temperature. Figure 4 shows typical stress vs. strain curves provided by tensile tests on CFRP laminates after curing. The curves are characterized by a linear elastic behavior, also in the larger displacement region until fracture, according to the results shown by Forcellese et al. [26] on unidirectional laminates in CFRP and by Di Bella et al. [22] on bidirectional-fabrics-reinforced composites.

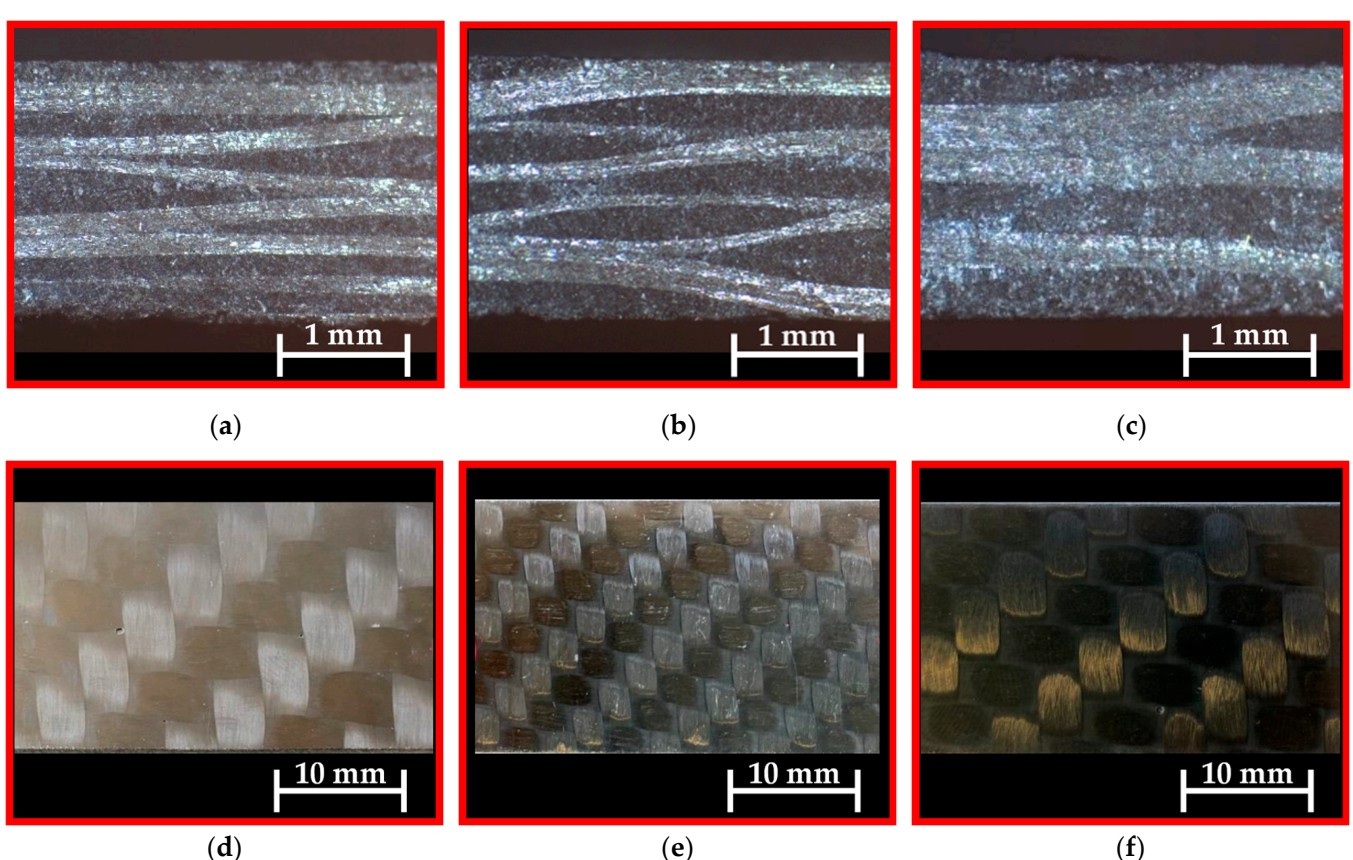

**Figure 3.** Initial microstructures of longitudinal sections and upper surfaces of composite laminates obtained using twill fabrics in CFRP with different AW: (**a**,**d**) 380 g/m$^2$; (**b**,**e**) 630 g/m$^2$; (**c**,**f**) 800 g/m$^2$.

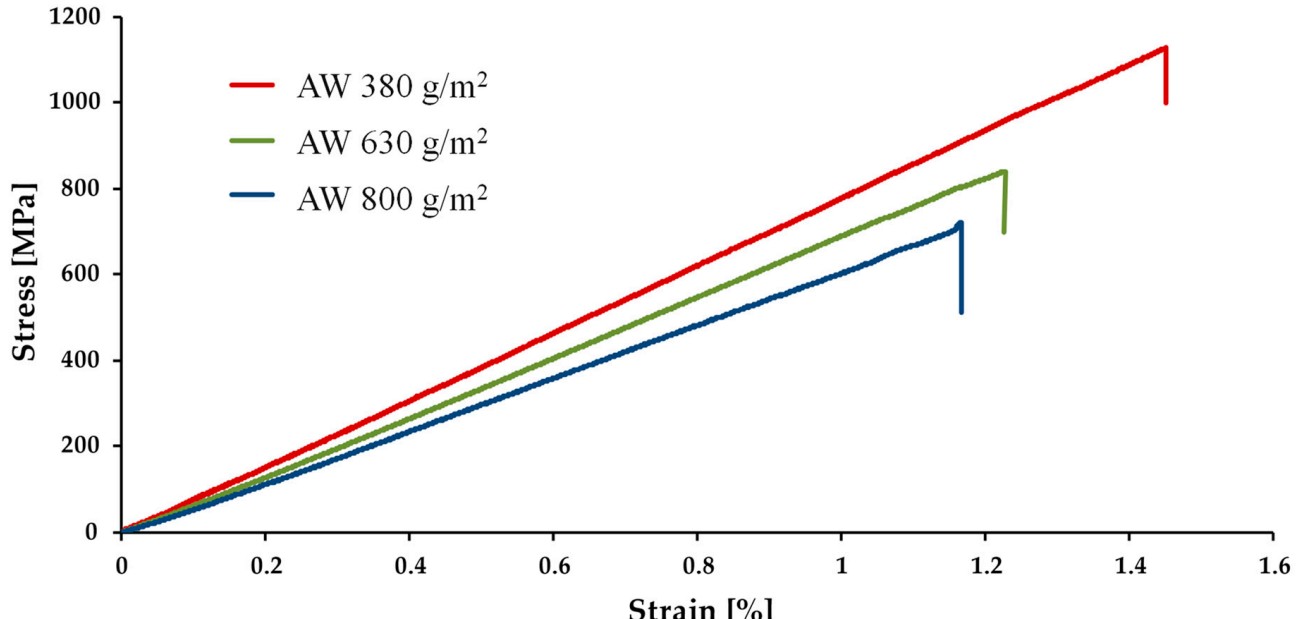

**Figure 4.** Effect of AW of twill fabric in carbon-fiber-reinforced composites on the normalized tensile stress vs. tensile strain curves.

As far as the fabric AW is concerned, it can be observed that, for a given strain level, a decrease in AW leads to an increase in strength and stiffness, according to the results

shown by Zhou et al. [4]. The average values of ultimate tensile strength, elastic modulus, and ultimate elongation are summarized in Table 3.

Fabrics with the lowest AW led to the obtaining laminates characterized by ultimate tensile strength, ultimate elongation, and elastic modulus values that are 34.7%, 15.8%, and 14.3%, respectively, higher than those realized using a fabric AW of 630 g/m$^2$, and 56.9%, 23.7%, and 26.6% higher, respectively, in respect of the fabric AW of 800 g/m$^2$. This means that the reduction in weight of fiber per unit area of the fabric significantly improves the load bearing capability and the strain-to-failure of the composite material.

**Table 3.** Normalized mechanical properties of composite materials from tensile tests and shear stress tests.

| Properties | AW 380 g/m$^2$ | AW 630 g/m$^2$ | AW 800 g/m$^2$ |
|---|---|---|---|
| Ultimate tensile strength [MPa] | 1134.1 ± 48.8 | 841.9 ± 27.9 | 722.7 ± 17.3 |
| Ultimate elongation [%] | 1.46 ± 0.07 | 1.26 ± 0.08 | 1.18 ± 0.05 |
| Elastic modulus [GPa] | 73.83 ± 1.98 | 64.57 ± 2.08 | 58.3 ± 2.75 |
| Maximum in-plane shear stress [MPa] | 82.83 ± 2.58 | 57.22 ± 0.46 | 53.3 ± 0.68 |
| In-plane chord modulus of elasticity [GPa] | 3.90 ± 0.25 | 3.97 ± 0.10 | 3.27 ± 0.07 |

The effect of fabric AW on the in-plain shear properties of laminates is similar to that observed for the tensile properties. To this purpose, Figure 5 shows typical shear stress vs. shear strain curves for the different fabric AW specimens.

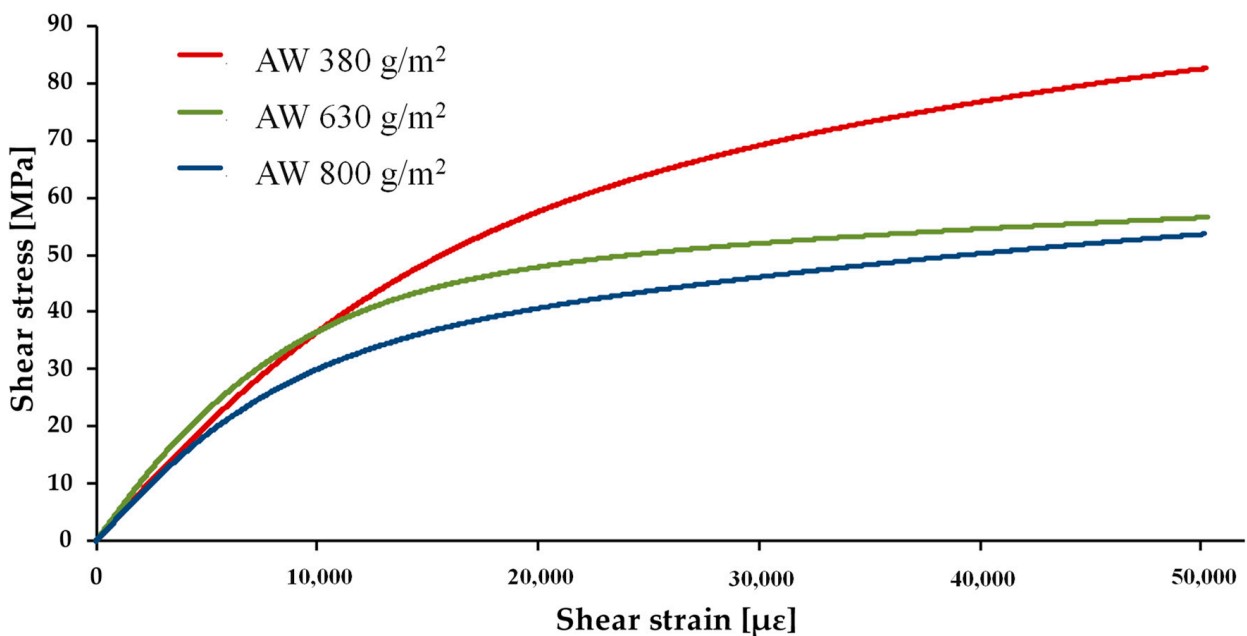

**Figure 5.** Effect of AW of twill fabric in carbon-fiber-reinforced composites on shear stress vs. shear strain curves.

It can be noted that the composite laminate manufactured by laying-up plies of 380 g/m$^2$ exhibits higher in-plane shear stress than those obtained using plies with an AW of 630 g/m$^2$ and 800 g/m$^2$; specifically, the laminate with the AW of 380 g/m$^2$ shows a maximum shear stress value 44.7% and 55.4% higher than that obtained using twill fabrics of 630 g/m$^2$ and 800 g/m$^2$. Such behavior can be attributed to the different fiber alignment. As a matter of fact, the fibers interweaving in the fabric with AW of 630 g/m$^2$ and 800 g/m$^2$ are characterized by more noticeable and pronounced undulations than those exhibited by fabric with AW of 380 g/m$^2$, as shown in Figure 3, resulting in different fabric mechanical behaviors. Instead, the in-plane shear chord modulus of elasticity is higher for the AW

of 630 g/m$^2$ than for the AW 380 g/m$^2$ and 800 g/m$^2$. As a matter of fact, for low shear strain values, the slope of the curve related to the AW of 630 g/m$^2$ is higher than that obtained for AW of 380 g/m$^2$. Such a slope tends to decrease as the shear strain increases; in particular, after a shear strain of 5000 $\mu\varepsilon$, the slope of the curve related to the fabric AW of 380 g/m$^2$ becomes higher than that at 630 g/m$^2$. On the other hand, the curve relative to the fabric of 800 g/m$^2$ is always below the other two curves, proving that it is characterized by values of in-plane shear stress and in-plane shear chord modulus lower than those of the other fabrics.

From the mechanical tensile and shear tests, it was highlighted, therefore, that the best behavior both in terms of mechanical tensile and shear strength, both in terms of ductility and stiffness, was obtained from samples realized using fabrics with AW equal to 380 g/m$^2$, as reported in Table 3, while the worst performances were obtained with samples with AW of 800 g/m$^2$.

Correlating the effect of the composite fabric areal weight on the mechanical properties of the laminates is crucial for an optimal material selection during the design process. Hence, these results can be immediately exploited in industrial applications.

Figure 6 is a representative scheme which explains the different crimp effect in the three analyzed fabrics. In this paper, the $2 \times 2$ twill involves weaving two bundles of weft fibers above and below two bundles of warp fibers. In both the fabric with a weight of 380 g/m$^2$ and 630 g/m$^2$, the bundles of fibers are 12K, but the difference lies in the shape of the section of the bundle itself. As can be observed in Figure 3d,e, the bundle width of the 380 g/m$^2$ fabric is wider than that of the 630 g/m$^2$. As a consequence, if $a_1 < a_2$ (Figure 5), the laminate with AW of 380 g/m$^2$ is characterized by a smaller bundle thickness ($b_1$) than the one in the laminate with AW of 630 g/m$^2$ ($b_2$). Such a result is in accordance with the cross-section shown in Figure 3a,b ($b_1 > b_2$). The same result can be obtained for the 800 g/m$^2$ fabric which has 24K fiber bundles. Comparing it to the 380 g/m$^2$ fabric, from Figure 3d,f, it is possible to notice that they have similar fiber bundle width ($a_1 \approx a_2$), both larger than the 630 g/m$^2$ fabric (Figure 3e). The difference is related to the thickness of the bundle which, being 24K instead of 12K, as can also be seen from Figure 3a,c, is approximately the double ($b_2 \approx 2b_1$). Therefore, as far as the 380 g/m$^2$ fabric is concerned, a pronounced fiber alignment can be observed, and the bundles of fibers follow almost linear trajectories as compared to the 630 g/m$^2$ and 800 g/m$^2$ fabrics. The difference among the mechanical performances of the different AW laminates is related to how the fibers are woven into the different fabrics.

As a matter of fact, if a thicker tow is used to produce to prepreg fabric, the crimp effect on the carbon fibers increases. This can reduce fiber alignment along typical loads directions, increasing possible stress concentrations and decreasing the overall part mechanical properties. On the other hand, if wider tows with lower thickness are used, the fibers appear more flattened and their alignment is improved, resulting in higher mechanical properties.

Figure 7 shows the $11\times$ magnification of the fracture zone of a typical specimen in composite laminate manufactured by laying-up plies of 380 g/m$^2$. In this case, the failure occurred for fracture of the fibers in a central area of the gauge length.

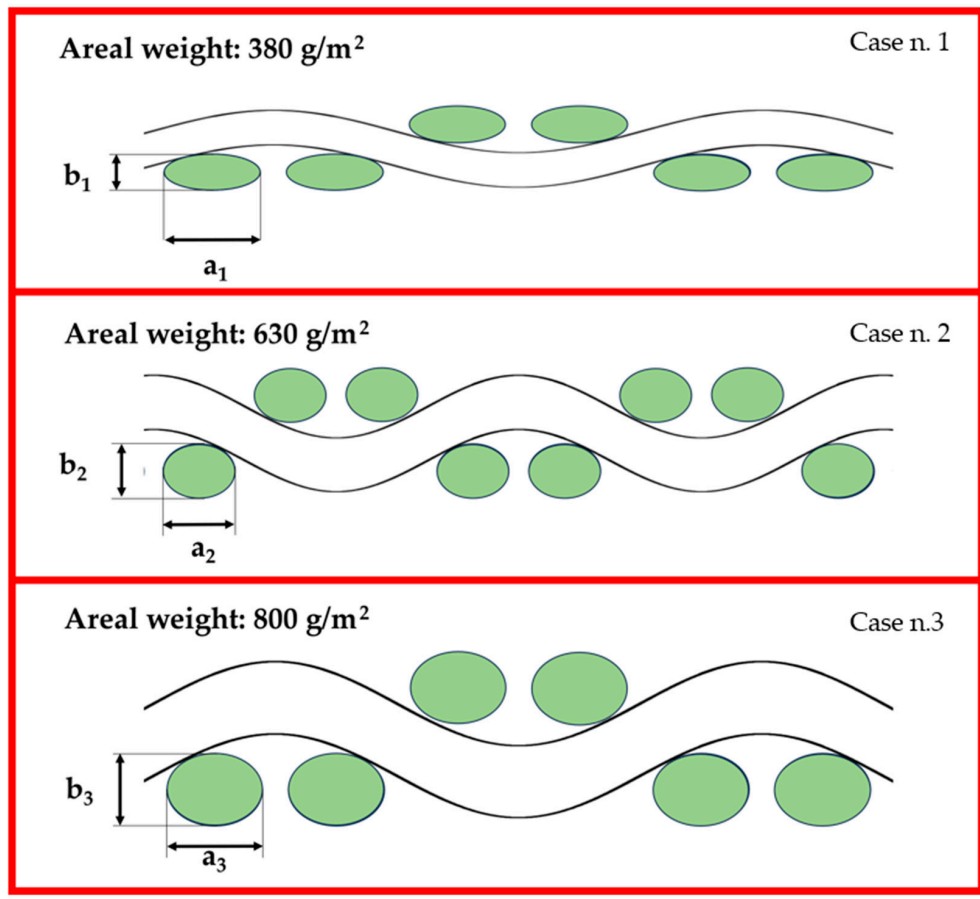

**Figure 6.** Schematic representation of the cross-section of twill fabric in carbon-fiber-reinforced composites at different AW (a = bundle width; b = bundle thickness).

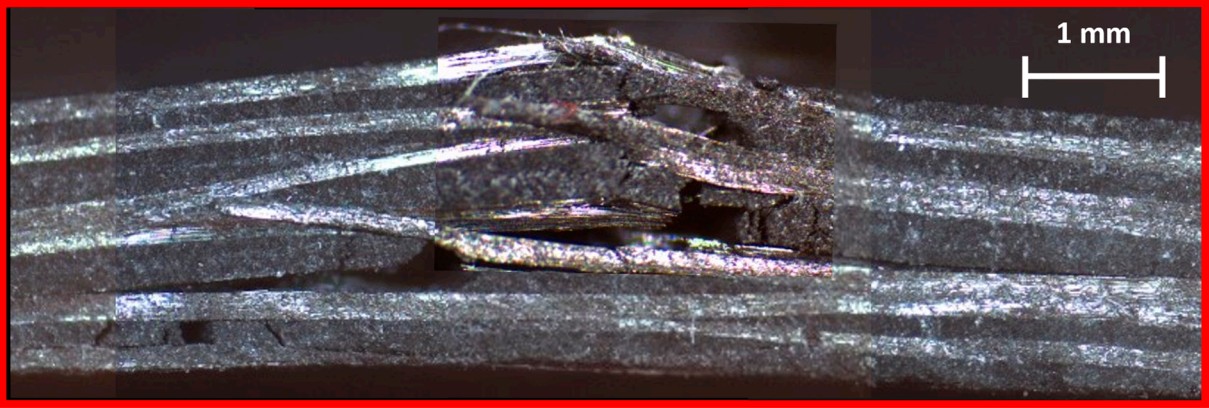

**Figure 7.** Fractured longitudinal section of a tensile specimen manufactured by laying-up plies of 380 g/m$^2$.

Figure 8 shows the 11× magnification of the fractured longitudinal section of a tensile specimen manufactured at an AW of 630 g/m$^2$. Unlike the specimen at AW of 380 g/m$^2$, the failure was caused by delamination, as plies were separated from each other in the perpendicular direction to that of the load application.

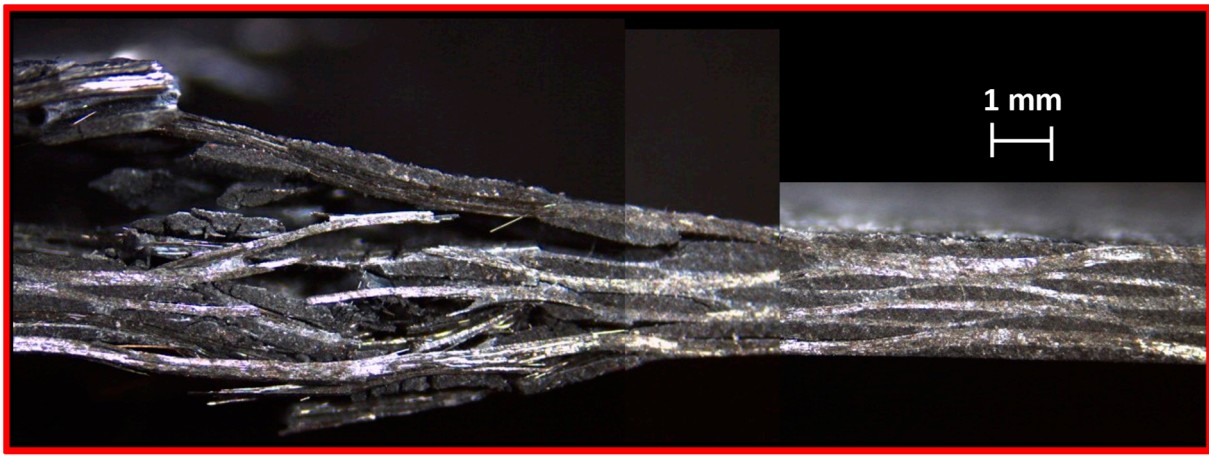

**Figure 8.** Fractured longitudinal section of a tensile specimen manufactured by laying-up plies of 630 g/m$^2$.

Figure 9 shows the fracture zone of a typical specimen in composite laminate manufactured by laying-up plies of 800 g/m$^2$. The failure mode of such a laminate is different from the one observed in Figure 8 since delamination is less marked and affects a very small area of the sample. Such behavior can be attributed to the size of the fibers which is equal to 24K for fabric with an AW of 800 g/m$^2$, whilst is 12K for fabric with an AW of 630 g/m$^2$. Every fracture mode is in accordance with that reported in ASTM D3039 in the conditions Lateral Gage Middle (LGM) for the 380 g/m$^2$ fabric and 800 g/m$^2$, and Delamination Gage Middle (DGM) for the 630 g/m$^2$ fabric.

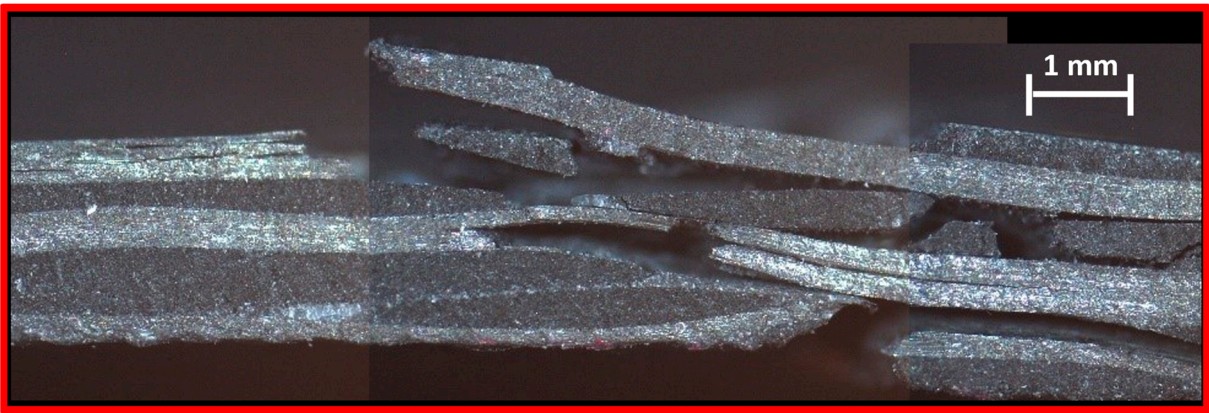

**Figure 9.** Fractured longitudinal section of the tensile specimen manufactured by laying-up plies of 800 g/m$^2$.

Figures 10–12 show the fractured longitudinal sections of the in-shear specimens in composite laminate manufactured by laying-up plies with AW of 380, 630, and 800 g/m$^2$, respectively. The specimen at the lowest AW investigated is characterized by a failure that occurs in a rather limited area of the specimen, with not very pronounced delamination (Figure 10). On the other hand, the specimen with AW of 630 g/m$^2$ exhibits a wider failure zone, in which delamination is predominant (Figure 11). The specimen realized with an AW of 800 g/m$^2$ is characterized by a failure zone with an extensive delamination (Figure 12). The difference in size for the carbon fiber tows used in specimens with AW of 800 g/m$^2$ (24K), as compared to the laminates with AW of 380 and 630 g/m$^2$ (12K), is clearly visible.

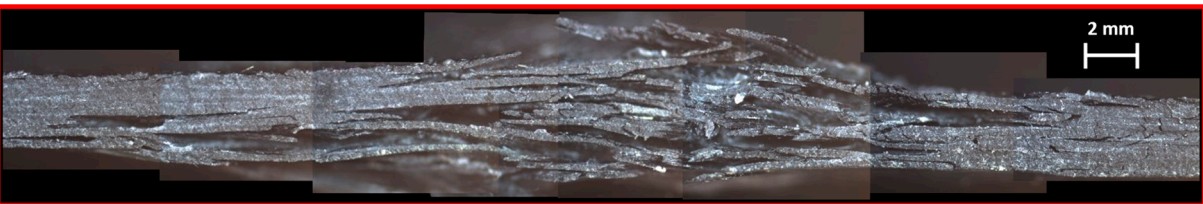

**Figure 10.** Fractured longitudinal section of the in-shear specimen in composite laminate manufactured by laying-up plies of 380 g/m$^2$.

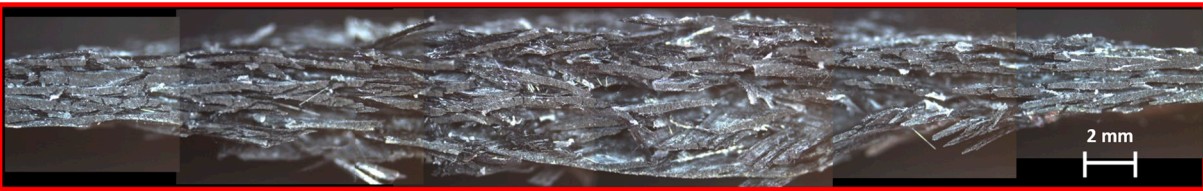

**Figure 11.** Fractured longitudinal section of the in-shear specimen in composite laminate manufactured by laying-up plies of 630 g/m$^2$.

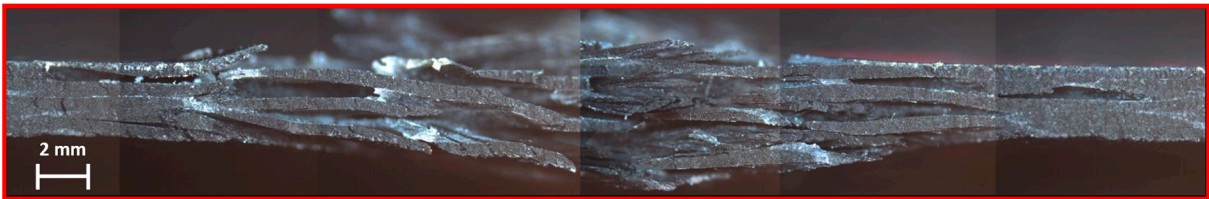

**Figure 12.** Fractured longitudinal section of the in-shear specimen in composite laminate manufactured by laying-up plies of 800 g/m$^2$.

Considering the marked delamination between the layers of towpreg observed in the above images, it appears that in some case the weight resin content is not adequate for the amount of reinforcement. This is the case of composite laminate manufactured by laying-up plies of 630 g/m$^2$ (resin content 32.1%$w/w$). This result can be attributed to the low resin content between the layers, which is inadequate to ensure the adhesion of the layers when the tensile and shear stresses reach values similar to the resistance to failure of the composite laminate. Therefore, a low amount of resin means that the sample failure occurs through a marked debonding of the towpreg layers. Similar considerations can be made for the fabric characterized by an AW of 800 g/m$^2$. As a matter of fact, even in this case the amount of resin is relatively low (29.7%) and leads to a break very similar to that of the fabric with an AW of 630 g/m$^2$. Unlike the last one, however, the size of the fibers is 24K instead of 12K, as can also be seen from Figures 11 and 12, in which the difference between the thicknesses of the fabrics is quite noticeable. However, when the value of the resin content rises, the debonding is greatly reduced as in the case of composite laminate manufactured by laying-up plies of 380 g/m$^2$ (resin content 43.9%$w/w$). This also increases the specimens' interlaminar shear strength. As a result, delamination takes place, and cracks propagate between the interfaces of the layers.

The results obtained from the tensile and in-plane shear stress tests showed that the different fabric AW affects the mechanical performances of the CFRP composite laminate. Such result can be attributed to the different compaction between overlying plies during vacuum bag operations and final autoclave curing process. Specifically, the lowest fabric AW leads to performing a more active compaction and, consequently, to reducing the void content in percentage in the final composite laminate, as also confirmed by the results of the digestion test (Table 2).

## 4. Conclusions

The effect of the fabric AW on the mechanical properties of composite laminates in carbon-fiber-reinforced polymers was investigated. Three pre-impregnated $2 \times 2$ twill weaves, characterized by different AW values of $380 \ g/m^2$, $630 \ g/m^2$, and $800 \ g/m^2$, were used to manufacture composite laminates. They were chosen due to their wide use in the automotive sector for producing structural components by exploiting their high drapability and limited crimping. The final thickness of the laminates was reached by laying-up a different number of plies, i.e., six layers with a fabric AW of $380 \ g/m^2$, four layers with an AW of $630 \ g/m^2$, and three layers with an AW of $800 \ g/m^2$. Uniaxial tensile and in-plane shear response tests were carried out on samples obtained from composite laminates.

Resin digestion tests were carried out to quantify the voids in the composite materials; finally, microscopy and stereomicroscopy analyses were employed to observe the plies and their cross-sections.

The main results can be summarized as follows:

- the void content in percentage is higher in the laminates obtained by laying-up fabrics with an AW of $630 \ g/m^2$ with respect to those obtained using the weaves with AWs of $380 \ g/m^2$ and $800 \ g/m^2$;
- for a given strain level, a decrease in the fabric AW leads to an increase in strength and stiffness;
- the twill waves specimens with the AWs of $380 \ g/m^2$ and $800 \ g/m^2$ subjected to tensile and in-plane shear tests demonstrate a less evident delamination in a limited failure zone, whilst the twill wave specimens with the AW of $630 \ g/m^2$ show a pronounced delamination phenomenon in a wider failure zone;
- the twill weave with an AW of $380 \ g/m^2$ is characterized by a maximum in-plane shear stress higher than those with an AWs of $630 \ g/m^2$ and $800 \ g/m^2$; on the contrary, the shear stress chord modulus is higher for the fabric with the AW of $630 \ g/m^2$ than for those with AWs of $380 \ g/m^2$ and $800 \ g/m^2$;
- the difference among the mechanical performances of the different AW laminates is related to how the fibers are woven into the different fabrics: as far as the $380 \ g/m^2$ fabric is concerned, a pronounced fiber alignment appears, and the bundles of fibers follow almost linear trajectories as compared to the $800 \ g/m^2$ fabrics.

These findings can help to better understand composite material behavior by linking their mechanical properties with laminates areal weight. Moreover, they can be employed for an optimal material selection during the design phase, with immediate industrial applications. For example, knowing that laminating a $630 \ g/m^2$ AW fabric results in obtaining a higher percentage of voids as compared to $380 \ g/m^2$ and $800 \ g/m^2$ AW fabrics can affect choices of laminates during the design phase of structural components.

In addition to these results, it is worth to notice that the number of plies has a great influence also on the manufacturing time and costs. This aspect must also be considered by the engineers during the design of CFRP components in order to find the best compromise between mechanical and manufacturing performances.

Further investigations will concern the evaluation of the mechanical performance of laminates realized with different tows (e.g., 3k, 6k, and 12k tows), different weave architectures (e.g., $2 \times 2$ and $4 \times 4$ twill), and consequent different AWs, with a particular focus on the fiber bundles analysis, both with optical microscopy and computerized tomography.

**Author Contributions:** Conceptualization, M.S., T.M. and I.B.; Methodology, M.S., T.M. and I.B.; Formal analysis, T.M. and I.B.; Investigation, M.A. and S.G.; Data curation, T.M.; Validation, I.B.; Project administration, M.S.; Supervision, M.S.; Writing—original draft, M.A., S.G. and I.B.; Writing—review & editing, T.M. and M.S. All authors have read and agreed to the published version of the manuscript.

**Funding:** The authors received no financial support for the research, authorship, and publication of this article.

**Data Availability Statement:** The data presented in this study are available on request from the corresponding author.

**Acknowledgments:** Authors acknowledge HP Composites S.p.a. for their collaboration in the experimental work.

**Conflicts of Interest:** The authors declared no potential conflict of interest with respect to the research, authorship, and/or publication of this article.

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
