# Peer review of "Effect of Fabric Areal Weight on the Mechanical Properties of Composite Laminates in Carbon-Fiber-Reinforced Polymers"

_jcs, doi:10.3390/jcs7090351_

Round 1

Reviewer 1 Report

This paper deals with investigation of mechanical properties of composite laminates in carbon fiber reinforced polymers.

This study will be worthy to be published on this journal after some corrections.

Some suggestions are as follows.

Comment 1: Figure 2 (b) and Figure 3

Are these figures needed?

Comment 2: Figure 4

The twill weave of (e) is the opposite direction with that of (d) and (f). Twill weave is usually diagonal designs from the top right corner to the lower left. Is the fabric (e) the back?

Author Response

Reviewer n.1

Comments and Suggestions for Authors

This paper deals with investigation of mechanical properties of composite laminates in carbon fiber reinforced polymers.

This study will be worthy to be published on this journal after some corrections.

Some suggestions are as follows.

Comment 1: Figure 2 (b) and Figure 3

Are these figures needed?

Figures 2b and 3 are not needed. For this reason, the authors have removed them from the manuscript. The numbering of the figures has been updated accordingly.

Comment 2: Figure 4

The twill weave of (e) is the opposite direction with that of (d) and (f). Twill weave is usually diagonal designs from the top right corner to the lower left. Is the fabric (e) the back?

Figure 4e (now Figure 3e) was accidentally flipped horizontally during the "collage" phase of the different images of the figure. The authors would like to thank the reviewer for noticing the error and allowing the authors to correct it under review.

Author Response

Reviewer n.2

  1. The objective is clearly stated, but it would be helpful if the authors elaborate on the significance of the research. Why is it crucial to understand the effect of the reinforcing fabric's areal weight on the mechanical properties of composite laminates?

Thank you for pointing that out. Indeed, correlating the effect of the composite fabric areal weight on the mechanical properties of laminates is crucial for an optimal material selection during the design process. Hence, this study has immediate industrial implications; moreover, it contributes to the scientific literature by helping understand the behaviour of composite materials and prepreg. These aspects were clarified in the paper.

  1. While the authors have mentioned the methods used in the study, some important details are missing. For example, what was the rationale behind choosing the three specific areal weights?

Thank you for the useful comment. In accordance with industrial experts' recommendations, the three different fabrics were selected to represent typical areal weight values used for structural applications. As a matter of fact, these fabrics are amongst the most used for automotive components production. This aspect was clarified in the abstract and in the methodology section.

  1. The results are summarized. However, the terminology and reporting of the results need to be more precise. For example, "significantly affects" is a rather vague term. It would be beneficial to provide the statistical significance or effect size.

Representative results data were added in the abstract.

  1. The language used in the abstract is generally clear and concise. There are some long sentences, which could be broken down into smaller sentences for better clarity.

Readability was enhanced according to the reviewer suggestion.

Section 1:

  1. The introduction provides a comprehensive background and states the purpose of the study. However, it is too long and could be more concise. Also, a few relevant references seem to be missing. For example, the statement "These fabrics are the most employed in the automotive sector" should be supported by a reference.

The introduction has been significantly changed and shortened, as suggested by the reviewer. Relevant references have been added.

  1. The authors have done a good job of explaining the context and providing a literature review. However, they should try to make the connections between the different concepts more explicit to improve the flow.

The flow has been improved by better connecting the concepts in section 1.

  1. The objective of the research is clearly stated, which is good. However, it would be helpful if the authors could explain more about why they selected the specific areal weight values for their study. The rationale behind the number of plies used for each fabric should be explained more clearly as well.

The authors explained more about why they selected the areal weight values investigated in their study. Furthermore, the motivation was better explained in the Materials and Methodology section.

  1. The structure of the introduction could be improved. The current structure seems to jump from one topic to another. It could start with a broad overview of the field, narrow down to specific problems or limitations, and then propose the study's objectives.

Thank you for the useful comment. The structure of the introduction was improved.

  1. The language used is generally good, but there are some long sentences that could be split to improve readability. There are also some minor typos that need to be corrected, such as "whit a section" should be "with a section".

The English was improved. Some typos were corrected.

  1. The relevance of the study to the existing literature and its potential contribution to the field could be emphasized more.

The relevance of the study to the existing liteature and its potential contribution to the field was emphasized more in section 1.

  1. In order to emphasize the importance of the subject the authors are encouraged to discuss pultruded FRP composites that are also widely used in automotive industry.

Please, refer to:

  1. https://doi.org/10.1016/j.compstruct.2022.116216
  2. https://doi.org/10.1177/00219983211001528

The authors believe that pultruded FRP is not relevant to the topic of the paper. However, the papers suggested by the reviewer have been included in the paper because the authors find them interesting.

Section 2:

  1. More background information about the choice of materials could strengthen the study. Why were these specific materials selected? What makes them suitable for this research? This will help readers understand the rationale behind the decisions.

Authors thank the reviewer for the comment. The authors improved the explanation on material selection for this research.

  1. This section is detailed and provides a clear understanding of the process. A brief description of the hand lay-up technique would be helpful for readers unfamiliar with the term.

Thank you for the suggestion. A brief description of the production process was added in Section 2.

  1. This section is detailed and well-structured. The manuscript provides clear details about the tests conducted to evaluate the composites. Including more about why these specific tests were chosen and what they aim to demonstrate would be helpful.

Authors thank the reviewer for the comment. The description of the aim for the choice of tensile and shear tests is implemented in the relative section.

Section 3:

  1. Ensure consistent formatting throughout the manuscript. For example, in some parts of the text, authors have written areal weight as 'AW', and in others, as 'areal weight'. It's best to maintain a consistent usage throughout to avoid any confusion.

Thank you for the useful comment. Authors modified the paper using only the 'AW' acronym instead of 'areal weight'.

  1. The manuscript could be improved by structuring it in a way that ensures the discussion follows a logical flow. The explanation of the results seems to jump back and forth between different properties and their relation to the fabric areal weight. Authors could group the discussion and conclusions about the same properties together for better coherence.

Authors thank the reviewer for the suggestion and confirm that the section has been modified in the following logical order for the presentation of the results: initial microstructure, tensile results, shear results, crimp effect and tensile and shear fracture magnification.

  1. In the context of tensile tests (lines 234-251), the discussion is thorough and provides a good comparative analysis. However, it might be useful to include a sentence or two explaining the implications of these results for potential real-world applications of these composite laminates.

Understanding the mechanical properties variation as a function of the fabric area weight is crucial for an optimal design process. This study allows a correct choice of material depending on the design requirements, with immediate industrial applications. This aspect was better clarified in Section 3 after the mechanical test results are presented.

  1. The analysis of fracture zones in lines 281-294 is detailed, but it could be improved by directly connecting each fracture zone to its corresponding fabric areal weight and weaving pattern.

Thank you for the comment. The fracture zones were highlighted in the last six figures of the manuscript and were described in relation to the different areal weights and weaving patterns.

  1. Authors mentioned "Figure 9Figure 9" on line 311, which seems to be a typo.

Authors corrected the typo in the manuscript.

  1. I appreciate that authors have tried to interpret and explain the obtained results. However, in some areas, the discussion could be more in-depth. For example, the analysis about how different AW values affect mechanical properties could be expanded.

Thank you for the suggestions. The discussion section was modified according to the reviewer suggestion to provide better explanation of the mechanical behaviour of the investigated laminates. More detailed comments of Figure 6 were added to better discuss the correlation between different AW values and the composite parts mechanical properties. Moreover, the results of the mechanical tests were moved close together to improve clarity and readability.

  1. The conclusions drawn from each set of results should be clearly stated before moving on to the next set. At times, it seems like the conclusions are being mentioned in passing without much emphasis.

Thank you for the suggestions. Related sections were implemented according to the reviewer suggestion to provide better explanation for each set of results. A summary sentence has been added at the end of the resin digestion analysis and the results of the mechanical tests.

Section 4:

  1. The authors effectively summarize the main findings in the introduction of the Conclusions section (lines 382-391). However, consider providing a more explicit statement on the significance of the study and its potential impact on the field.

Thank you for the suggestion. As stated in the previous sections, this study has relevant scientific and industrial impacts since it allows to better understand composite materials behaviour and supports material selection in industrial design. An explicit statement was added in the final part of the results section.

  1. Consider explaining more about why these findings matter. For example, why does it matter that the void content in percentage is higher in the laminates obtained by laying-up fabrics with an areal weight of 630 g/m2?

Thank you for the useful comment. Authors modified the conclusions section emphasizing the importance of the obtained results, adding a specific section after the bulleted.

  1. The final statement about future investigations (lines 415-416) gives a good sense of direction for subsequent research. However, consider specifying what particular mechanical performance or weave architecture aspects will be investigated, as it could help readers understand the future focus and potential significance of the study.

Thank you for the comment. The results section was modified according to the reviewer suggestion by adding specific data about future work. The study will continue with an in dept assessment of the mechanical properties of different fabrics used in industrial applications; some examples include 4x4 fabric twill realized with different size tows (3k, 6k, 12k).

Round 2

Reviewer 1 Report

This article was revised enough, so it can be published.

Reviewer 2 Report

All major comments were adequately addressed and the Authors have done an admirable job of improving the quality of the manuscript. Therefore, it can be accepted without any structural modification.